# An Efficient Algorithm for Infrared Earth Sensor with a Large Field of View

**DOI:** 10.3390/s22239409

**Published:** 2022-12-02

**Authors:** Bendong Wang, Hao Wang, Zhonghe Jin

**Affiliations:** Micro-Satellite Research Center, School of Aeronautics and Astronautics, Zhejiang University, Hangzhou 310027, China

**Keywords:** infrared Earth sensor, modified RANSAC, weighted total least squares

## Abstract

Infrared Earth sensors with large-field-of-view (FOV) cameras are widely used in low-Earth-orbit satellites. To improve the accuracy and speed of Earth sensors, an algorithm based on modified random sample consensus (RANSAC) and weighted total least squares (WTLS) is proposed. Firstly, the modified RANSAC with a pre-verification step was used to remove the noisy points efficiently. Then, the Earth’s oblateness was taken into consideration and the Earth’s horizon was projected onto a unit sphere as a three-dimensional (3D) curve. Finally, the TLS and WTLS were used to fit the projection of the Earth horizon. With the help of TLS and WTLS, the accuracy of the Earth sensor was greatly improved. Simulated images and on-orbit infrared images obtained via the satellite Tianping-2B were used to assess the performance of the algorithm. The experimental results demonstrate that the method outperforms RANSAC, M-estimator sample consensus (MLESAC), and Hough transformation in terms of speed. The accuracy of the algorithm for nadir estimation is approximately $0.04^{{}^{\circ}}$ (root-mean-square error) when Earth is fully visible and $0.16^{{}^{\circ}}$ when the off-nadir angle is $120^{{}^{\circ}}$, which is a significant improvement upon other nadir estimation algorithms

## 1. Introduction

An accurate and reliable attitude determination system is required for most spacecraft missions. Attitude determination systems usually consist of multiple sensors, such as a star tracker, infrared Earth sensor, sun sensor, magnetometer, etc. [1,2]. Among them, infrared Earth sensors are widely used in spacecraft for their reliability and low cost [3].

Infrared Earth sensors can be roughly divided into two categories: scanning Earth sensors and static Earth sensors. The scanning Earth sensors rotate along a certain axis and detect infrared signals from Earth. Then, the roll angle of the spacecraft will be calculated based on the time that the Earth runs across the sensor’s range of view. Some examples include STD-15 [4], STD-16 [5], and IERS [6]. Although this kind of Earth sensors could achieve a high accuracy, they are not well suited for small satellites because of their larger size and greater power consumption [7].

On the other hand, traditional static Earth sensors use multiple thermopile detectors to detect the Earth horizon and calculate the spacecraft’s attitude. These include MAI-SES, manufactured by Maryland Aerospace [3], and HSNS, manufactured by Solar MEMS Technologies [8]. The MAI-SES’s accuracy is $0.18^{{}^{\circ}}$(position known) while the HSNS’s accuracy is better than $1^{{}^{\circ}}\left(3\sigma \right)$. Generally, traditional static Earth sensors are low-cost and easy to implement, but their performance is affected by the variation of radiance in the atmosphere [9].

In recent years, many static Earth sensors use 2-D image sensors instead of thermopile detectors. The basic idea is to capture the infrared images of the Earth and extract horizon points with image-processing algorithms. Then, the Earth horizon points are fitted to a circle, ellipse, parabola, or hyperbola and the nadir vector will be obtained from the parameters of the conic [10]. For example, the accuracy of the digital Earth sensor designed by SITAEL is better than $1^{{}^{\circ}}$ and the Fov is $37^{{}^{\circ}}\times 44^{{}^{\circ}}$ (horizontal and vertical) [11]. Saadat used a miniature vehicle inspection camera as an Earth sensor and the accuracy was $0.69^{{}^{\circ}}\left(1\sigma \right)$ [12]. Kikuya et al. proposed a three-axis attitude determination algorithm using a visible-ray camera with a resolution of 3280×2464 pixels as an Earth sensor [13]. With the help of image identification technology, this Earth sensor’s accuracy was similar to a coarse sun sensor. Recently, Modenini took the Earth oblateness into account and proposed a solution to estimate the attitude from imaged elilipsoids [14]. The accuracy of the roll and pitch determination achieved $0.01^{{}^{\circ}}$. The performance of the prototype was not reported when the off-nadir angle was over $10^{{}^{\circ}}$. Furthermore, the sensor prototype was designed with three different cameras to capture the entire Earth limb, which may introduce additional error. In [15], Christian modified Modenini’s solution and proposed a tutorial to infer the relative position, attitude, or both from the observed body’s horizon. This comprehensive work developed different algorithms for different optical navigation scenarios. For the spacecraft venturing in low-Earth orbit (LEO), many Earth sensors use wide FOV cameras, such as cameras with fisheye lens [16]. The most representative commercial sensor of this kind is the CubeSense manufactured by CubeSpace [17]. Its accuracy is up to $0.2^{{}^{\circ}}\left(3\sigma \right)$ if the Earth is fully visible and the FOV is $130^{{}^{\circ}}$ (horizontal and vertical). However, the effect of Earth oblateness is not considered and the accuracy decreases when the Earth moves towards the edge of the FOV. Barf designed a horizon sensor with a fisheye camera for sounding rockets [18,19]. Its error is below $0.5^{{}^{\circ}}$. Two Cubesats, Athenoxat-1 and ISARA, also used fisheye cameras to obtain nadir vector, but no further information about the sensors they used is available [20,21].

In our previous work [22], an Earth sensor composed of a panoramic annular lens (PAL) and a complementary-metal-oxide-semiconductor (CMOS) infrared camera was designed. The Earth sensor has been used on multiple missions [23]. Compared with the Earth sensors using pinhole cameras, it only needs one camera to capture the entire Earth horizon. It can also operate in low-Earth orbit (LEO) satellites and estimate the nadir vector even when the off-nadir angle is over $90^{{}^{\circ}}$. Similarly to the CubeSense, the accuracy decreases as the off-nadir angle increases and it did not take the Earth oblateness into consideration. Additionally, Hough transformation is used to remove outliers, which is robust but time-consuming.

In order to solve these problems, an algorithm based on modified RANSAC and WTLS is proposed. The basic idea is to use modified RANSAC to remove the disturbance points, then the weights of the Earth edge points are calculated. After that, the TLS and WTLS technique is applied to fit the Earth edge points to a three-dimensional(3D) curve instead of a conic section. Compared with the conventional Hough transformation and MLESAC algorithm, the time consumption is dramatically reduced. Furthermore, the accuracy of the sensor is especially improved when the Earth limb is small.

The main contributions of this paper are:A modified RANSAC with a pre-verification procedure is used to remove outliers. A small amount of data instead of all measured data are used to qualify the established models, which is the pre-verification procedure that improves the efficiency.The Earth horizon points are mapped onto the unit sphere instead of the image plane, which forms a three-dimensional curve instead of a conic section. The 3D curve fitting is more robust than conic fitting for PAL images or fisheye images.The WTLS is introduced into the 3D curve fitting which is different for each horizon point’s precision. Consequently, the accuracy of the sensor is improved.

The rest of this paper is organized as follows: In Section 2, the proposed method is described by covering the modified RANSAC, the expression for the projection of the Earth horizon points, and WTLS-based 3D curve fitting. In Section 3, the implementation and performance of the proposed algorithm are denoted. In Section 4, conclusions and future work are presented.

## 2. Algorithm Description

The proposed algorithm consists of five steps. Firstly, the edge detection algorithm is performed to extract edge points in the image. Secondly, the edge points are mapped onto the unit image-sphere. Thirdly, the modified RANSAC is performed to remove the disturbance points. Fourthly, the actual horizon is fitted to a 3D curve with WTLS. Finally, the Nadir vector is calculated with the coefficients of the curve. The flowchart of the entire algorithm is shown in Figure 1.

### 2.1. Edge Detection

There are several kinds of edge detection algorithms that can be applied to the Earth horizon detection, such as Sobel, Prewitt, and Canny [24]. To increase the accuracy, we used the subpixel edge-localization algorithm developed by Christian [25,26]. Sobel edge detection was used to find the approximate edge location. Then, the subpixel edge localization algorithm using Zernike moments was performed to find the accurate edge points in the image. In this paper, the size of the Zernike masks is 7×7.

### 2.2. Horizon Projection

The lens follows the principle of the flat-cylinder perspective [27]. The FOV of the PAL camera we used is $360^{{}^{\circ}}$ in azimuth and $90^{{}^{\circ}}$ in zenith. The imaging formula of PAL is given by:(1)r=fθ
where *f* is the focal length of the PAL, *r* is the distance between the object and the lens’ center, and θ is the angle from the optical axis. According to [15], the Earth horizon in the image plane, i.e., the z=1 plane, must be a conic. Most algorithms project edge points onto the image plane [28,29,30]. Set the coordinates of an observed point in the camera frame as $\left[X_{C},Y_{C},Z_{C}\right]$, then the image plane coordinates of the observed point $\left[x_{z},y_{z}\right]$ are given by:(2)$$\begin{array}{cc} x_{z} & =\frac{X_{C}}{Z_{C}}\\ y_{z} & =\frac{Y_{C}}{Z_{C}} \end{array}$$

For an observed point near the edge of FOV, its $Z_{C}$ is close to 0. As a result, its image plane coordinates will be large and inaccurate, which would lead the conic fitting process to fail. Thus, projecting the edge points onto the image plane and fitting them to a conic is not suitable for a wide-FOV camera such as PAL. In order to solve this problem, the edge points are mapped one-to-one onto the unit image-sphere, just like our previous work. The mapping is defined as follows:(3)$$\begin{array}{cc} x & =\sin\theta_{i}\frac{u_{i}}{\sqrt{u_{i}^{2}+v_{i}^{2}}}\\ y & =\sin\theta_{i}\frac{v_{i}}{\sqrt{u_{i}^{2}+v_{i}^{2}}}\\ z & =\cos\theta_{i} \end{array}$$
where [x,y,z] are the coordinates of the point on the sphere, and $\left[u_{i},v_{i}\right]$ are the corresponding image coordinates. The angle $\theta_{i}$ is determined from Equation (Equation 1). After the projection, the Earth horizon on the image-sphere is a 3D curve instead of a conic, and the details shall be found in Section 2.4.

### 2.3. Modified RANSAC

The extracted edge points may not only contain the actual Earth horizon, but also the boundary between areas with different infrared radiance. There are many algorithms that can be applied to remove the disturbance points: among them RANSAC algorithms are widely used [31,32,33]. The classic RANSAC algorithm randomly selects sample points to establish a model, which is often an error model due to the disturbance points. Thus, all other points have to be checked if the model was rejected. This means that it is significantly time consuming. Although the Earth horizon on the image-sphere is a 3D curve, it is also close to a circle because the Earth’s ellipticity is neglectable. Hence, this information can be used to verify the model. The basic flow of the modified RANSAC is as follows:Randomly select five sample points. The coordinates of the *i*-th point are $\left[x^{i},y^{i},z^{i}\right]$.Calculate the normal vector to the plane determined by three of them using Equation (Equation 4):
(4)$$\begin{array}{cc} p & =[x^{1}-x^{2},y^{1}-y^{2},z^{1}-z^{2}]\\ q & =[x^{1}-x^{3},y^{1}-y^{3},z^{1}-z^{3}]\\ n & =\vec{p}\times \vec{q} \end{array}$$Calculate the angles between the normal vector and the vectors pointing from the origin to the sample points, respectively, $\theta_{1}$ – $\theta_{5}$. $\theta_{m}$ is the mean of these angles. If the mean deviation A.D.>T, go back to step 1.Calculate the angles between the normal vector and vectors pointing from the origin to the rest points, for instance, $\theta_{i}$, if $\left|\theta_{i}-\theta_{m}\right|<T$, the point is considered as an inlier. The number of inliers is denoted as $N_{k}$.If $N_{k}>N_{max}$, then set $N_{max}=N_{k}$.Repeat steps 1–5 $k_{max}$ times. Note that, if a set with 50% inliers is discovered, end the loop.Remove all the outliers and extract the actual horizon. Furthermore, the normal vector n is approximately the nadir vector and thus the approximate off-nadir angle φ can be obtained.

### 2.4. Three-Dimensional Curve Fitting

As mentioned above, the projection of the Earth horizon on the unit sphere is a 3D curve. In this section, the representation of the curve is derived and the fitting method is described.

#### 2.4.1. Projection of Earth Horizon on the Unit Sphere

The geometry of the Earth sensor and Earth is shown in Figure 2.

As shown in Figure 2, $O_{C}$ is the origin of the camera coordinate system, while $O_{E}$ is the Earth’s center. $S_{i}$ is the vector pointing from $O_{C}$ to an Earth horizon point. r is a vector pointing from $O_{E}$ to $O_{C}$. The shape matrix of the Earth $A_{p}$ is calculated as:(5)Ap=11Ra2Ra200011Rb2Rb200011Rc2Rc2
where $R_{a}=R_{b}=6378.137$ km and $R_{c}=6356.755$ km [34,35]. According to [15], the vector from the camera origin to the Earth horizon point in the camera frame, $S_{i}$, obeys the constraint:(6)$${S_{i}}^{T}MS_{i}=0$$
Normalized $S_{i}$ is the projection of a horizon point on a unit sphere. Furthermore, M is given by:(7)$$\begin{array}{cc} M & =Arr^{T}A-\left(r^{T}Ar-1\right)A\\ A & =T_{c}^{p}A_{p}T_{p}^{c} \end{array}$$
$T_{c}^{p}$ denotes the rotation matrix from an Earth coordinate system to the Earth sensor’s coordinate system and $T_{p}^{c}$ is the transpose of $T_{c}^{p}$. Since M is symmetric, it can be written as:(8)M=abb22dd22bb22cee22dd22ee22f
Moreover, Equation (Equation 6) is rewritten as
(9)$$ax^{2}+bxy+cy^{2}+dxz+eyz+fz^{2}=0$$
where *x*, *y*, and *z* are the coordinates of the normalized $S_{i}$. This is the representation of the projection of Earth horizon on the unit sphere as well as a 3D curve. Since $x^{2}+y^{2}+z^{2}=1$, Equation (Equation 9) is the same as
(10)$$\left(1-\frac{a}{f}\right)x^{2}-\frac{b}{f}xy+\left(1-\frac{c}{f}\right)y^{2}-\frac{d}{f}xz-\frac{e}{f}yz=1$$

#### 2.4.2. Weighted Total Least Squares

Define ξ as
(11)$$\xi ={\left[\xi_{1},\xi_{2},\xi_{3},\xi_{4},\xi_{5}\right]}^{T}={\left[\left(1-\frac{a}{f}\right),-\frac{b}{f},\left(1-\frac{c}{f}\right),-\frac{d}{f},-\frac{e}{f}\right]}^{T}$$Let $\left[x_{i},y_{i},z_{i}\right]$ denote the *i*-th edge point’s coordinates, define $e_{i}$ as
(12)$$e_{i}=\left[x_{i}^{2},x_{i}y_{i},y_{i}^{2},x_{i}z_{i},y_{i}z_{i}\right]$$A linear system can be built based on Equation (Equation 10)
(13)$$E\xi =1_{n\times 1}$$
where
(14)$$E=\left[\begin{array}{c} e_{1}\\ e_{2}\\ \vdots \\ e_{n} \end{array}\right]$$
Since E is noisy, it is more suitable to use the total least squares to estimate ξ [36,37]. Furthermore, the precision of each point’s coordinates is different. Thus, the weighted total least squares can achieve better accuracy. However, the weighted total least squares requires more time. Therefore, the total least squares is used to estimate the initial value of ξ. Then, the weighted total least squares is only performed when the off-nadir angle is large, because the coordinates of the points near the edge of FOV are less accurate.

The singular value decomposition (SVD) is used to find the solution to the total least squares problem. Define matrix Ey as
(15)$$\mathrm{Ey}=\left[\begin{array}{cc} E & 1_{n\times 1} \end{array}\right]$$
Then, singular value decomposition of Ey is performed:(16)$$\mathrm{Ey}=U\sum V^{T}$$
The estimation of ξ is given by:(17)$$\xi =-\frac{V(1:5,6)}{V(6,6)}$$

The weighted total least squares will be performed when the approximate off-nadir angle φ is greater than $\varphi_{T}$. The cofactor matrices of the design matrix and observation vector are denoted by $Q_{E}$ and $Q_{y}$. $Q_{y}$ is set as a zero matrix because the observation vector is a constant vector.

Substituting Equation (Equation 3) into Equation (Equation 12), the functional relationship between the elements of a design matrix and the coordinates of the edge points on the image is given by: (18)$$\begin{array}{cc} e_{i} & =\left[x_{i}^{2},x_{i}y_{i},y_{i}^{2},x_{i}z_{i},y_{i}z_{i}\right]\\ \; & =\left[\sin^{2}\theta_{i}\frac{u_{i}^{2}}{u_{i}^{2}+v_{i}^{2}},\sin^{2}\theta_{i}\frac{u_{i}v_{i}}{u_{i}^{2}+v_{i}^{2}},\sin^{2}\theta_{i}\frac{v_{i}^{2}}{u_{i}^{2}+v_{i}^{2}},\sin\theta_{i}\cos\theta_{i}\frac{u_{i}}{\sqrt{u_{i}^{2}+v_{i}^{2}}},\sin\theta_{i}\cos\theta_{i}\frac{v_{i}}{\sqrt{u_{i}^{2}+v_{i}^{2}}}\right] \end{array}$$

Symbolize the function between $e_{i}$ and $\left[u_{i},v_{i}\right]$ as *F*. The functional relationship between E and $\left[u_{i},v_{i}\right]$ is given by:(19)$$vec\left(E\right)=\left[\begin{array}{c} F_{1}\left(u_{1},v_{1},u_{2},v_{2},\cdots u_{n},v_{n}\right)\\ F_{2}\left(u_{1},v_{1},u_{2},v_{2},\cdots u_{n},v_{n}\right)\\ \vdots \\ F_{5n}\left(u_{1},v_{1},u_{2},v_{2},\cdots u_{n},v_{n}\right) \end{array}\right]$$
where vec denotes the operator that stacks one column of a matrix underneath the previous one. According to the law of the propagation of cofactors [38], $Q_{E}$ is calculated by
(20)$$Q_{E}=J_{uv}Q_{uv}J_{uv}^{T}$$
where $J_{uv}$ is a matrix of partial derivatives
(21)$$J_{uv}=\left[\begin{array}{cccc} \frac{\partial F_{1}}{\partial u_{1}} & \frac{\partial F_{1}}{\partial v_{1}} & \cdots & \frac{\partial F_{1}}{\partial v_{n}}\\ \frac{\partial F_{2}}{\partial u_{1}} & \frac{\partial F_{2}}{\partial v_{1}} & \cdots & \frac{\partial F_{2}}{\partial v_{n}}\\ \vdots & & & \vdots \\ \frac{\partial F_{5n}}{\partial u_{1}} & \frac{\partial F_{5n}}{\partial v_{1}} & \cdots & \frac{\partial F_{5n}}{\partial v_{n}} \end{array}\right]$$
Assuming that the precision of observables $u_{i}$ and $v_{1}$ is the same, then $Q_{uv}$ is an identity matrix.

Once the cofactor matrix is obtained, the parameters can be estimated according to [39]. The pseudocode is listed as follows:
$\xi_{0}$ is estimated from TLSfor i=1 to N do
 $\xi_{i-1}^{{}^{\prime }}=\xi {}_{i-1}⊗I_{n}$ $Q_{1}^{i}={\xi_{i-1}^{{}^{\prime }}}^{T}Q_{E}\xi_{i-1}^{{}^{\prime }}$ $vec\left(\delta_{E}^{i}\right)=-Q_{E}\xi_{i-1}^{{}^{\prime }}{\left(Q_{1}^{i}\right)}^{-1}\left(1_{n\times 1}-E\xi {}_{i-1}\right)$ $\delta_{E}^{i}=\mathrm{reshape}\left[vec\left(\delta_{E}^{i}\right)\right]$ $E_{i}=E-\delta_{E}^{i}$ $\xi {}_{i}={\left({E_{i}}^{T}{\left(Q_{1}^{i}\right)}^{-1}E_{i}\right)}^{-1}{E_{i}}^{T}{\left(Q_{1}^{i}\right)}^{-1}\left(1_{n\times 1}-\delta_{E}^{i}\xi {}_{i-1}\right)$end when $∥\xi {}_{i}-\xi {}_{i-1}∥<\varepsilon$

Once ξ is obtained, set *f* to 1, then M is given by:(22)$$M=\left[\begin{array}{ccc} 1-\xi {}_{1} & -\frac{\xi {}_{2}}{2} & -\frac{\xi {}_{4}}{2}\\ -\frac{\xi {}_{2}}{2} & 1-\xi {}_{3} & -\frac{\xi {}_{5}}{2}\\ -\frac{\xi {}_{4}}{2} & -\frac{\xi {}_{5}}{2} & 1 \end{array}\right]$$

With M, the nadir vector can be estimated by Christian’s algorithm, the details of which can be seen in [15].

## 3. Experiments

### 3.1. Calibration of PAL

Real lenses do not exactly follow the designed projection model. Additionally, the origin of the Earth sensor coordinate needs to be determined. Thus, it is better to calibrate the PAL before using it. The PAL lens’ projection model is given by [40]
(23)$$r\left(\theta \right)=k_{1}\theta +k_{2}\theta^{3}+k_{3}\theta^{5}+\ldots$$
The calibration devices mainly include a two-axis rotator and an infrared source. The details of the calibration method can be found in [41].

### 3.2. Simulation System

The proposed algorithm was verified and compared with other studies on a series of simulated Earth images and real images. The parameters of the optical system are shown in Table 1.

The Earth infrared radiance is effected by latitude, season, and atmospheric pressure, etc. [42]. The radiance of the Northern Hemisphere increases with increasing latitude in the summer and decreases with latitude in the winter. The radiance also decreases with increasing tangent height. The geometry of the tangent height is shown in Figure 3.

Ref. [43] provided the relation curves of the Earth infrared radiance at different latitudes as a function of the tangent height $H_{T}$. The latitudes in [43] were set from $15^{{}^{\circ}}$ N to $90^{{}^{\circ}}$ N with a step of $15^{{}^{\circ}}$, the seasons was summer and winter, and tangent height was from −30 km to 70 km. We assumed that the radiance of the Northern Hemisphere in summer is the same as that of the Southern Hemisphere in winter and the radiance at any latitude can be calculated by the known profiles using linear interpolation. For example, the radiance at $20^{{}^{\circ}}$ N can be calculated by P15+(P30−P15)×(20−15)(20−15)(30−15)(30−15), where $P_{15}$ and $P_{30}$ denote the radiance at $15^{{}^{\circ}}$ and $30^{{}^{\circ}}$, respectively. The relationship between the image intensity and the radiance can be determined according to on-orbit images. Given a point at equator ($0^{{}^{\circ}}$ N, 0 km tangent height), its radiance is denoted by $P_{0}$, then the corresponding gray value is set as the mean gray value of the Earth image $I_{e}$. The gray value of background noise is set as $I_{b}$. Then, the relationship between THE image intensity and radiance is given by:(24)$$I_{n}=\frac{I_{e}-I_{b}}{P_{0}}P_{n}+I_{b}$$
The Earth images were simulated as follows:Step1: Randomly set the Earth sensor’s position and attitude.Step2: For each pixel of the image sensor, the line of sight is set as the vector from the camera origin to the pixel’s projection on the unit-image sphere. Calculate the corresponding tangent height and latitude.Step3: Calculate each pixel’s radiance with tangent height $H_{T}$ and latitude.Step4: The image intensity of each pixel is calculated by Equation (Equation 24).Step5: Blur the image by a Gaussian function to simulate the effect of defocusing. Then, add Gaussian noise to the image.Step6: Add noisy points to the edge points to simulate the effect of clouds.

Figure 4a,b represent a simulated Earth image without noisy points and a real Earth image obtained by the Tianping-2B satellite, respectively. Note that the real Earth image has been cropped.

## 4. Results

### 4.1. Computational Efficiency

The modified RANSAC was compared with MLESAC, conventional RANSAC, and Hough transformation on simulated images. These algorithms were implemented in MATLAB on PC with Intel Core 3.10 GHz. The iteration number $k_{max}$ can be calculated with
(25)$$k_{max}=\frac{\log(1-p_{success})}{\log(1-{\left(1-r_{out}\right)}^{3})}$$
where $p_{success}$ is the probability that the actual horizon is extracted and $r_{out}$ is the ratio of the outliers. In our simulation, $p_{success}$ was set as 90% and $r_{out}$ was set as 80%, which is the worst case we assumed; then, the iteration number $k_{max}$ was set as 286. The threshold of the modified RANSAC *T* was set as $1^{{}^{\circ}}$. The ratio of noisy points was set from 10% to 70%.

The running times of these algorithms are shown in Figure 5. As can be seen, the Hough transformation is slower than other algorithms. Because it needs to go through the whole parameter space to find the optimal solution. The modified RANSAC is faster than MLESAC and conventional RANSAC because the wrong models are eliminated by the sample points.

The success rate of these algorithms is shown in Figure 6. As can be seen, the Hough transformation is more robust than other algorithms, and its success rate remains over 99%. However, the modified RANSAC can still achieve a success rate of 97.2% when the ratio of the outliers is 80%, and it can be improved by increasing the iteration number.

### 4.2. Accuracy

The proposed nadir vector estimation algorithm (TLS and WTLS) was tested on simulated images along with the circle fitting algorithm [22] and non-iterative nadir vector estimation algorithm [19]. The orbit altitude was set to 600 km. The off-nadir angle was set from $0^{{}^{\circ}}$ to $120^{{}^{\circ}}$ with a step of $0.1^{{}^{\circ}}$. For each off-nadir angle, 500 images were generated. The rotation matrix and latitude of the Earth sensor in these images were randomly set.

The nadir vector error represents the angular separation between the estimated nadir vector and the true nadir vector. The root mean square (RMS) of the nadir vector error is shown in Figure 7 and Figure 8. As can be seen, when the off-nadir angle is below $90^{{}^{\circ}}$, using TLS and WTLS to fit the Earth horizon can achieve nearly the same accuracy. Their RMS errors (RMSEs) are below $0.1^{{}^{\circ}}$. When the off-nadir angle is below $25^{{}^{\circ}}$, which is the scenario in which the Earth is fully visible, their RMSEs are approximately $0.04^{{}^{\circ}}$. As for the circle-fitting algorithm and non-iterative algorithm, their RMSEs are over $0.1^{{}^{\circ}}$. This is mainly because these algorithms did not consider the Earth’s oblateness. The Earth’s oblateness may cause a maximum error of $0.19^{{}^{\circ}}$. When the off-nadir angle is over $90^{{}^{\circ}}$, the RMSE of TLS fitting increases from $0.1^{{}^{\circ}}$ to $0.27^{{}^{\circ}}$, while the RMSE of WTLS fitting increases from $0.1^{{}^{\circ}}$ to $0.16^{{}^{\circ}}$. Thus, our algorithm uses WTLS to fit the Earth horizon when the off-nadir angle is over $90^{{}^{\circ}}$. As for the circle fitting algorithm and non-iterative nadir vector estimation algorithm, their RMSEs are approximately $0.14^{{}^{\circ}}$ and $0.15^{{}^{\circ}}$, respectively. However, the accuracy of these algorithms would be worse when the latitude is near $45^{{}^{\circ}}$.

Compared with Cubesense, whose accuracy is $0.2^{{}^{\circ}}\left(3\sigma \right)$ when Earth is fully visible, the accuracy of our algorithm is similar, however, the Earth’s oblateness is considered in our algorithm and the resolution of our Cmos is lower. Furthermore, our algorithm is more accurate when the off-nadir angle is large. Furthermore, compared with another commercial sensor MAI-SES whose accuracy is $0.18^{{}^{\circ}}$ (RMSE), our algorithm outperforms.

### 4.3. Performance on Real Earth Images

The Earth sensor was carried on the Tianping-2B satellite, which was developed by Zhejiang University and launched in March 2022. Several images of Earth were taken in orbit, as shown in Figure 9 and Figure 10. As shown in Figure 9c and Figure 10c, the red lines on the sphere are the 3D curves formed by Earth horizon points. Because the attitude matrix of the satellite is unknown when taking pictures, it is difficult to evaluate the accuracy of the proposed algorithm. However, the efficiency of the algorithm can be evaluated. Table 2 shows the time for the three algorithms to remove outliers. As can be seen, the proposed algorithm is more efficient.

## 5. Conclusions

A nadir vector estimation algorithm combining a modified RANSAC and TLS is proposed in this paper. It was intended to improve the efficiency and accuracy of Earth sensors with large-wide-FOV cameras. The algorithm uses a modified RANSAC algorithm to remove the noisy points and projects the Earth horizon onto the unit sphere as a 3D curve instead of a conic, which is more suitable for large-wide-FOV cameras. Then, the TLS and WTLS techniques are used to fit the 3D curve and improve the accuracy. The algorithm also takes the Earth’s oblateness into consideration. The experiments show that this algorithm is efficient and the accuracy is $0.04^{{}^{\circ}}$ when the Earth is fully visible. The RMSE increases as the off-nadir angle increases, but it can still achieve an accuracy of $0.16^{{}^{\circ}}$ when the off-nadir angle is $120^{{}^{\circ}}$. Furthermore, the proposed algorithm is much faster in dealing with real pictures taken in orbit than those of the Hough transformation and MLEASAC. Future work will focus on further improving the accuracy and validating the proposed algorithm to be used in an Earth sensor.

## Figures and Tables

**Figure 1 sensors-22-09409-f001:**
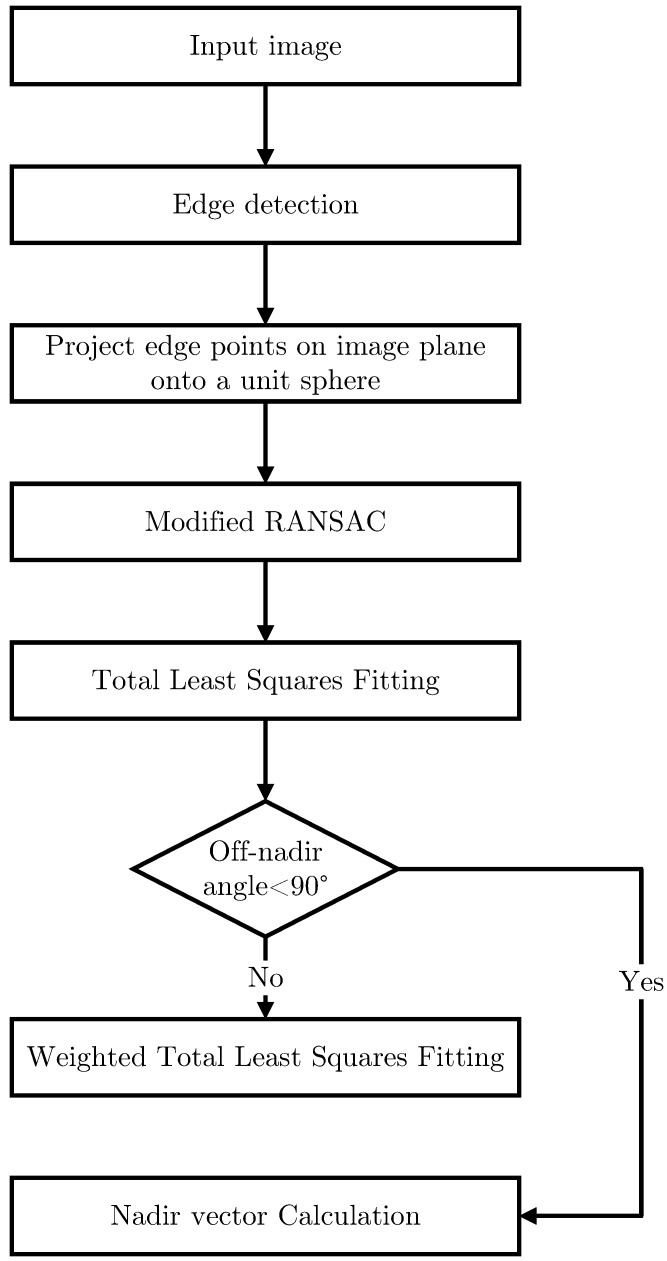
Flowchart of the proposed algorithm.

**Figure 2 sensors-22-09409-f002:**
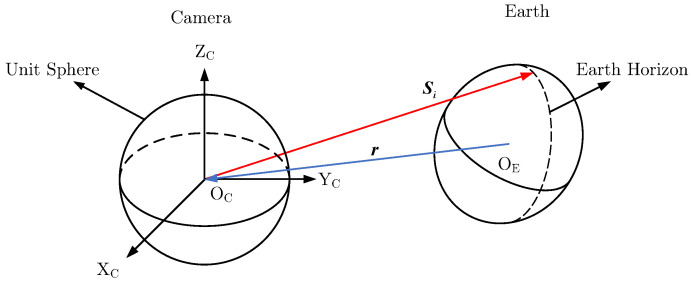
The geometry of Earth sensor and Earth.

**Figure 3 sensors-22-09409-f003:**
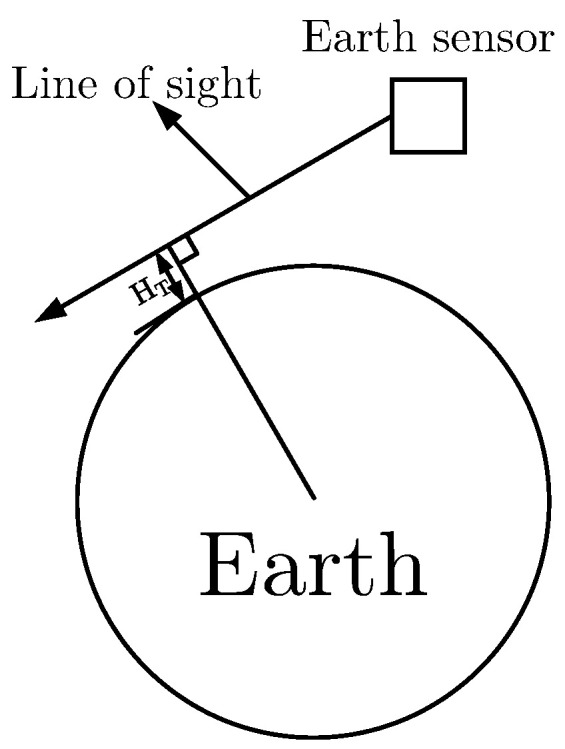
Geometry of tangent height.

**Figure 4 sensors-22-09409-f004:**
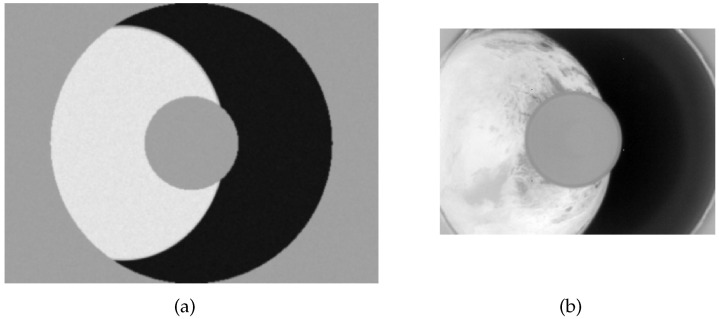
(**a**) Simulated Earth image without noisy points; and (**b**) Real Earth image.

**Figure 5 sensors-22-09409-f005:**
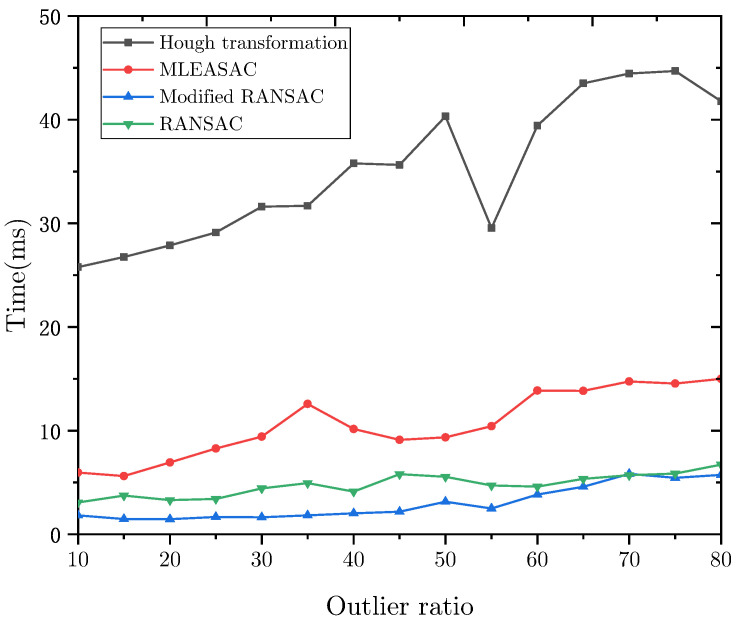
Time consumption vs. the ratio of noisy points.

**Figure 6 sensors-22-09409-f006:**
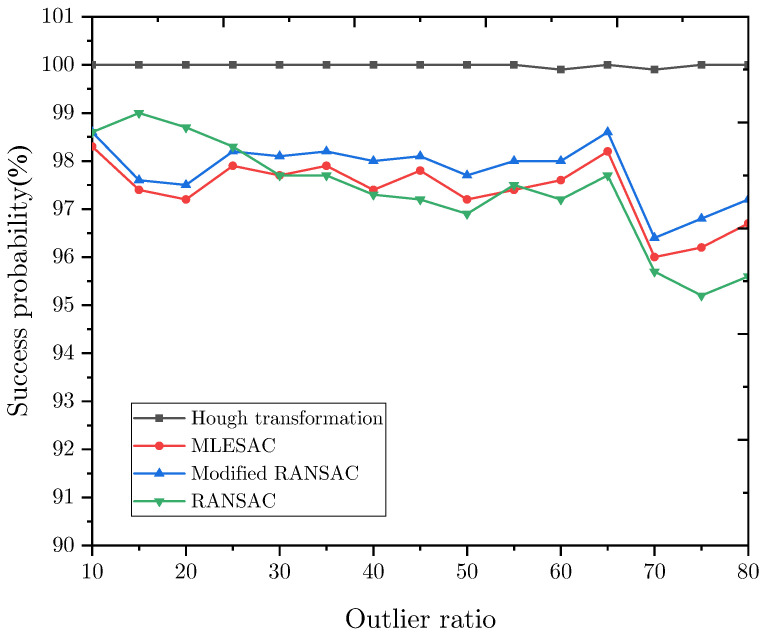
Success probability vs. the ratio of noisy points.

**Figure 7 sensors-22-09409-f007:**
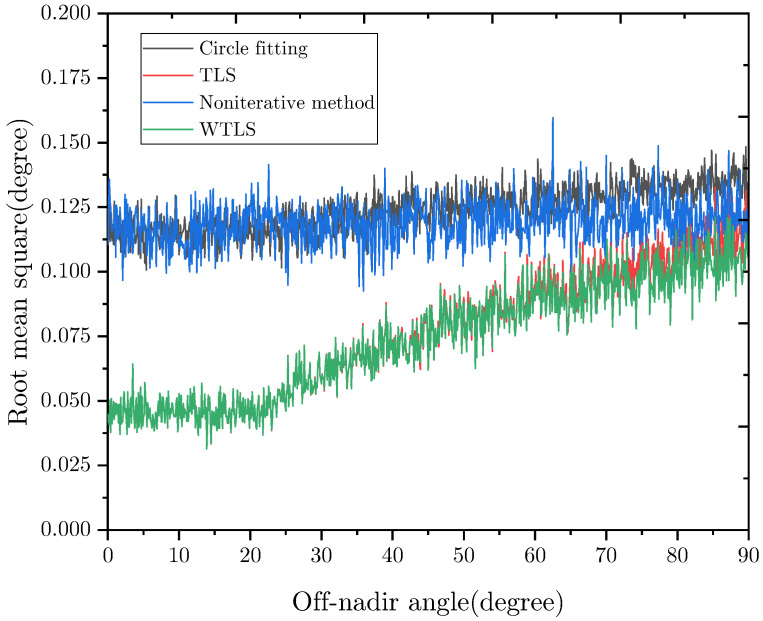
RMS error of the nadir vector vs. off-nadir angle. Off-nadir angle = 0–$90^{{}^{\circ}}$.

**Figure 8 sensors-22-09409-f008:**
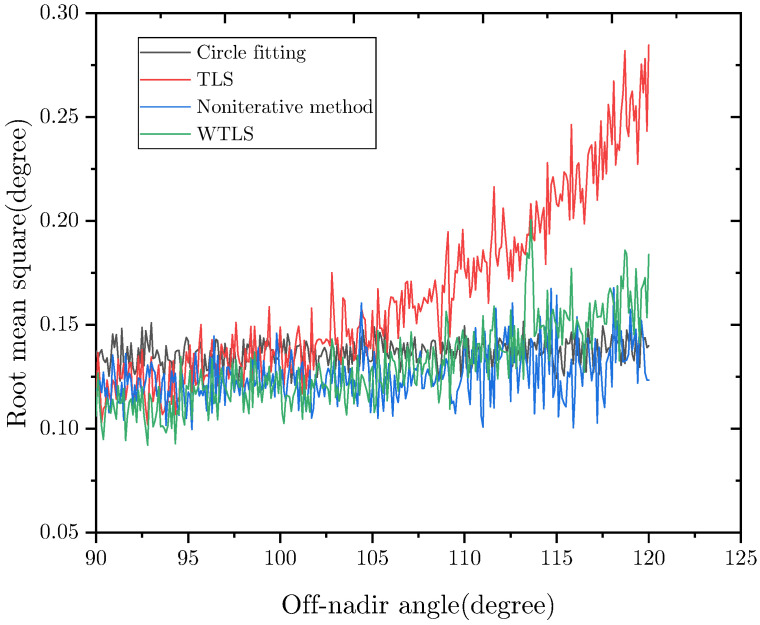
RMS error of the nadir vector vs. off-nadir angle. Off-nadir angle = 90–$120^{{}^{\circ}}$.

**Figure 9 sensors-22-09409-f009:**
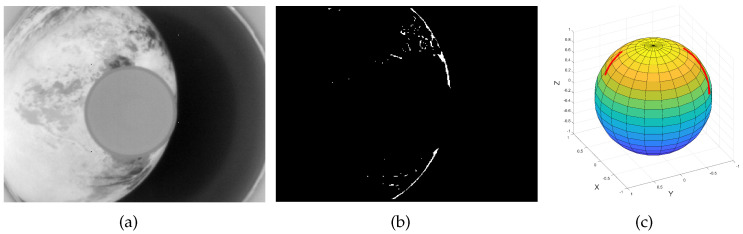
(**a**) Real Earth image. (**b**) Edge detected. (**c**) Actual horizon extracted.

**Figure 10 sensors-22-09409-f010:**
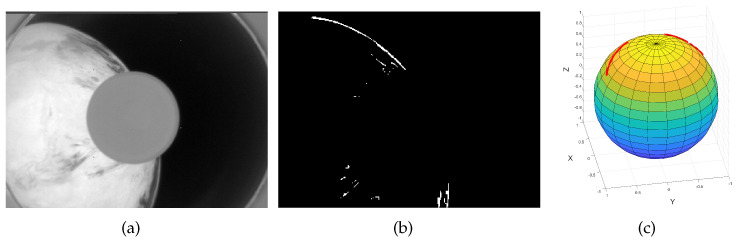
(**a**) Real Earth image. (**b**) Edge detected. (**c**) Actual horizon extracted.

**Table 1 sensors-22-09409-t001:** Specifications of the infrared camera.

Parameter	Value
**Resolution of CMOS**	288×384
**Size of one pixel**	25μm×25μm
**Focal length**	2.3mm
**FOV**	$180^{{}^{\circ}}\times 360^{{}^{\circ}}$
**Spectral range**	8−14 μm
**Dimension**	Φ40×30
**Weight**	40 g

**Table 2 sensors-22-09409-t002:** The time for outlier detection.

Algorithm	Figure 8	Figure 9
**Proposed algorithm**	8.89 ms	13.84 ms
**Hough transformation**	35.66	36.13 ms
**MLESAC**	27.94 ms	33.36 ms

## Data Availability

Not applicable.

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
