# Peer review of "An Efficient Algorithm for Infrared Earth Sensor with a Large Field of View"

_sensors, 2022, doi:10.3390/s22239409_

Round 1
Reviewer 1 Report
This manuscript proposes a new nadir vector estimation algorithm combines a modified RANSAC and TLS for infrared Earth sensor applications. Before proceeding to publication stage, please address my following comments:
1. Please provide references and source of constant values R_a, R_b, and R_c in equation (5)
2. There are many English issue throughout the manuscript, for example at [Line 147] "Since E is noisy, it is attractive to use the total least squares to estimate ξ". "Attractive" is not a good use of word here. Please replace with other word.
3. [Line 196] Please consideder change the subtitle of Section 3.2.1. from "Time Performance" to something else such as "Computational Efficiency".
4. [Line 233-235] "Because the attitude matrix of satellite is unknown when taking pictures, it’s hard to evaluate the accuracy of the proposed algorithm." - Can you compare your result against the result using other algorithms for benchmarking? Accuracy evaluation of real images is critical to support your conclusion.
Reviewer 2 Report
In my opinion, the paper is well-written and interseting. Maybe, the author can update the recent reference to improve the introduction part. I have find some papers for for your consideration:
1. H. Wang "A versatile method for target area coverage analysis with arbitrary satellite attitude maneuver paths," Acta Astronaut., vol. 194, pp. 242-254. 2022
2. X. Wang, "Agile Earth Observation Satellite Scheduling Over 20 Years: Formulations, Methods, and Future Directions," in IEEE Systems Journal, vol. 15, no. 3, pp. 3881-3892
Author Response
We thank the reviewer’s advice. The recommended references were cited.
Reviewer 3 Report
Good morning,
thank you for the paper. The report is attached. I would suggest to correct the editorials and maybe to extend the Conclusions and put it to the context of other works that is clear where your work belongs comparing to others. If somebody would be just interested in Abstract and conclusions, there is not enough information wo work with.
Regards,
reviewer

Reviewer 4 Report
Major comment
1. The improvement of the modified RANSAC can be stated more clearly in Section 2.3. As in lines 79-81, modified RANSAC has the pre-validation procedure.
2. Also in the results (Table 2), only the results for Hough transformation and MLESAC are used for comparison; one would naturally expect how the proposed algorithm compares with the unmodified RANSAC.
3. In addition to the time efficiency of the outlier removal algorithms, it should be important to compare the performance in terms of the accuracy of the determined nadir angle. Since one algorithm may remove points faster than the others but may yield worse determination accuracy due to removing wrong points. Section 3.2.1 seems to suggest an answer but it is unclear how many “correct” noisy points were removed from all three algorithms.
4. Figures 8 and 9, real earth images are incorrectly put with the wrong “edge detected” figures
Minor comment
1. PAL is undefined in the text.
2. Table 1, FOV of 180x360 does not make sense.
Reviewer 5 Report
An algorithm based on modified random sample consensus and weighted TLS is presented in the paper.
This research aims to solve the problem described in the introduction well.
Chapter 2 presents the mathematical method in a well-written manner.
In Chapter 2, the experiment and results are presented. In my opinion, these two paragraphs should be separated and the results analysis should be expanded. What is the computational cost of this method? Do you think this can be solved near real-time? These algorithms have been tested on a computer with 3.1GHz running Matlab. What can be done to overcome this limitation that limits further application of an in-space solution?
Could you provide other examples with images taken from the Tianping-2B satellite where the attitude matrix is known so that the processed results can be compared to the actual data? otherwise as sentences "it's hard to evaluate the accuracy of the proposed algorithm"
Round 2
Reviewer 5 Report
Ready to be accepted